# Effect of Grass Buffer Strips on Nitrogen and Phosphorus Removal from Paddy Runoff and Its Optimum Widths

Kexin Miao [1,2], Wanqing Dai [3], Zijian Xie [1,2,*], Chunhua Li [1,2] and Chun Ye [1,2]

1  State Environmental Protection Key Laboratory for Lake Pollution Control, Chinese Research Academy of Environmental Sciences, Beijing 100012, China; kexinmiao@gmail.com (K.M.); lich@craes.org.cn (C.L.)
2  National Engineering Laboratory for Lake Pollution Control and Ecological Restoration, Chinese Research Academy of Environmental Sciences, Beijing 100012, China
3  Department of Applied Science, School of Science and Technology, Hong Kong Metropolitan University, Good Shepherd Street, Ho Man Tin, Hong Kong SAR, China
*  Correspondence: zjxie2016@163.com; Tel./Fax: +86-010-84915191

**Abstract:** Paddy runoff pollution is one of the major contributors to limiting the improvement of water quality in Taihu Lake Basin. Grass buffer strips (GBSs) are an effective measure to control paddy runoff pollution. However, most studies only consider a single inflow condition, and few studies have considered the effect of high-frequency rainfall. In this study, a field runoff simulation experiment was constructed to simulate the effect of GBSs on runoff nitrogen and phosphorus removal at different inflow volumes, inflow velocities, inflow concentrations, and rainfall frequencies. Results demonstrated that the larger the inflow volume, the faster the inflow velocity, and the lower the inflow concentration, the higher the runoff pollutant interception rate that occurred in GBSs, and the interception rate improved significantly with increasing GBS widths. The peak change point of removal rate occurred at a width of 15 m for $NO_3^-$-N and TP and at a 25 m width for TN and $NH_4^+$-N. The cumulative removal rate increased slowly after the change point. Although the peak cumulative removal rate appeared at a GBS width of 35~45 m. Considering the pollutants intercepted by GBSs and the emerging demand for land in this basin, 25 m was recommended as the optimum width to remove runoff pollutants.

**Keywords:** non-point source pollution; paddy runoff; grass buffer strips; pollutant removal rate; optimum width

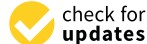



## 1. Background

Non-point source pollution is the primary source of water environment pollution when point source pollution has been effectively controlled [1,2]. Agricultural non-point source pollution (AGNPS) is the most critical contributor to watershed non-point source pollution [2–4]. The Yangtze River is the longest river in China. Previous research indicated that almost 50% of the total nitrogen (TN) and total phosphorus (TP) input to the river comes from AGNPS [5]. Similarly, regarding the second longest river of the Yellow River basin, TN and TP from agricultural sources contribute more than 40% and 60% of the watershed pollution load, respectively [6]. Taihu Lake is the third-largest freshwater lake in China. The water environment of Taihu Lake began to deteriorate in the 1990s, with the water quality decreasing from Class I and II to Class V based on the Environmental quality standards for surface water in China [7,8]. The main pollution factors were TN and TP [8–10]. Although the treatment of Taihu Lake has achieved some success in recent years, the problem of water eutrophication is still severe [7,10]. AGNPS is the main pollution source of Taihu Lake Basin, with a TN contribution of over 50% and a TP contribution of 30~40% [9]. Taihu Lake basin is one of the broadest rice-intensive planting areas, with 75% of the total cultivated land planted with rice [7,10,11]. In addition, the annual precipitation in most areas of the watershed exceeded 1000 mm and concentrated in the rainy season [10,11]. The

high frequency of rainfall in the watershed may result in higher runoff losses [7,11]. Since the rainy season coincides with the rice planting season in this region, runoff caused by rainfall carries large amounts of nitrogen and phosphorus from paddy fields, which could lead to a decrease in the water quality in Taihu Lake [7,11,12]. Therefore, how to effectively control runoff pollution from paddy fields is an essential issue to improve the water quality of Taihu Lake.

A vegetative buffer strip is an ecological engineering measure that has been proven to effectively alleviate water pollution in protected areas by preventing the migration and transport of pollutants through the synergistic action of vegetation, microorganisms, and soil [12–14]. The ability of vegetation buffer strips to mitigate pollutants is influenced by several factors, such as buffer strip width, inflow contaminant concentration, inflow velocity, etc. [14–17]. Bu et al. [18] confirmed that the hybrid vegetation buffer zone could intercept 74.4–86.3% of silt, 54.5–68.1% of TN, and 62.3–77.5% of TP. Lee et al. [19] found that under natural rainfall conditions, a 7.1 m buffer zone could intercept 80% of TN and 62% of $NO_3^-$-N, while a 16.3 m buffer zone could intercept 94% of TN and 85% of $NO_3^-$-N. She et al. [20] researched the influence of inflow velocity and inflow contaminant concentration on the physical blockage and mechanical filtration of plant stems. Results indicated that with lower discharge velocity, the simulated vegetation buffer zone showed a better performance of a retarding effect, but the inflow contaminant concentration had an insignificant effect on the vegetation buffer zone interception performance [20]. Additionally, Li et al. [21] confirmed that under the same inflow velocity, the Bermuda grass buffer zone's removal efficiency of TN, $NH_4^+$-N, and TP is negatively correlated with the inflow volume. It is worth mentioning that the above studies were mainly based on the results of a single inflow condition simulation, and few studies have focused on the effects of buffer strips in removing runoff pollutants under the influence of high-frequency rainfall in the Taihu Lake Basin. Thus, it is necessary to explore the effect of vegetation buffer strips on runoff pollutant removal under various inflow conditions and high-frequency rainfall.

This study selected the grass buffer strips (GBSs) to target the common problem of nitrogen and phosphorus pollution from paddy runoff in the Taihu Lake basin. A field simulation experiment was performed, while the main inflow conditions were inflow volume, inflow velocity, and inflow concentration. Meanwhile, simulations were also performed for the amount of nitrogen and phosphorus removed by GBSs under high-frequency rainfall. Then, the equation between the GBS width and pollutant removal rate was fitted, and the optimal width of GBSs for contaminant removal was explored. The results are intended to provide a scientific basis for controlling runoff contaminants in paddy fields and improving the water environmental quality in the middle and lower reaches of the Yangtze River.

## 2. Materials and Methods

### 2.1. Study Area

The experimental site is located in Wuxi City, Jiangsu Province, China, on the northern side of Taihu Lake. It is one of the typical rice-intensive growing areas in the middle and lower Yangtze River basin. The region belongs to the subtropical humid monsoon climate zone, with an average annual temperature of 16.2 °C and precipitation of 1121.7 mm. The flood season is concentrated from May to September. The average number of rainy days during the flood season is approximately 104, which accounts for 56.2~75.8% of the flood season.

The soil in the experiment site has 33.3~38.7% of clay, 42.7~49% of silt, and 13.1~13.7% of sand. The GBSs are dominated by Bermuda grass and have a coverage of 90%. A small number of crabgrass and barnyard grasses are distributed. The physical and chemical properties of the experiment site soil are given in Table 1.

**Table 1.** Soil physical and chemical properties in the field trial.

| Depth (cm) | pH (-) | Bulk Density (g/cm$^3$) | Organic Matter (g/kg) | Total N | NH$_4^+$-N (mg/kg) | NO$_3^-$-N | Total P | CaCl$_2$-P |
|---|---|---|---|---|---|---|---|---|
| 0–5 | 6.77 | 0.98 | 35.07 | 1.10 | 25.86 | 2.57 | 487.08 | 2.54 |

### 2.2. Simulation Experiment Design and Sample Collection

The experimental platform consists of a water distribution bucket, inlet and outlet pipes, a steady flow channel, and a GBS. A PE bucket is used for water storage. A submerged pump with a flow rate of 10 m$^3$/h is placed in each bucket. A pipe connects the bucket, valve, and flow meter to regulate the velocity of water flow. Water enters a steady flow channel to smooth the flow after discharge to avoid concentrated flows.

The GBS has a slope of 2°~3°, with a 2 m width and a 45 m length. It is separated by PVC plates, with a 20 cm buried depth of the baffle to prevent the mutual interference of runoffs. The native soil is below the GBS. Runoff samples were collected at 5 m, 15 m, 25 m, 35 m, and 45 m distances to the water inlet in each GBS.

Moreover, considering that this region has 20~30 heavy rainfalls during the rainy season, the experiment was performed ten consecutive times in a row every five days to simulate the high-frequency rainfall runoff. A total of 500 mL of water samples were collected at each runoff sampler, and three parallel samples were taken for laboratory analysis.

### 2.3. Inflow Condition Setting

Based on previous studies on the characteristics of rain runoff pollution in paddy fields, two concentrations of runoff were identified, denoted as high concentration and low concentration [14,22,23]. TN concentration was 7 mg/L and 12 mg/L, NH4$^+$-N was 5 mg/L and 9 mg/L, NO$_3^-$-N was 2 mg/L and 3 mg/L, and TP was 0.4 mg/L and 0.8 mg/L. Runoff water was configured with ammonia chloride, potassium nitrate, and superphosphate (Sinopharm Chemical Reagent Co. Ltd., Shanghai, China) through manual allocation.

Experimental runoff allocation and distribution time were calculated based on Wuxi's average rainfall and rainfall time in the flow season (May to September) in the last five years. Considering that the average rainfall of multi-year rainstorms in the region from 2015 to 2019 is 35.9 mm, the rainfall in this study was 36 mm. The coefficient of flow production was set to be 0.54, with 45 min to 90 min rainfall duration and 2 t to 4 t experimental discharge volume. The simulated paddy field covered about 200 m$^2$. The experiment GBS width was 2 m, with an inflow velocity ranging from 0.8 to 8 L/s. According to the experiment conditions, the inflow velocity was set to be 0.74 L/s and 1.48 L/s. Four groups of experiments were designed (Table 2).

**Table 2.** Simulation experiment scheme.

| Number | Inflow Volume (t) | Time (min) | Inflow Velocity (L/s) | Inflow Concentration |
|---|---|---|---|---|
| S1 | 2 | 45 | 0.74 | Low concentration |
| S2 | 4 | 45 | 1.48 | Low concentration |
| S3 | 4 | 90 | 0.74 | Low concentration |
| S4 | 4 | 90 | 0.74 | High concentration |

### 2.4. Sample Analysis

TN in runoff was determined using the Alkaline potassium persulfate digestion UV spectrophotometric method [24]. TP was determined using the Ammonium molybdate spectrophotometric method [25]. Ammonia nitrogen (NH$_4^+$-N) was determined through Nessler's reagent spectrophotometry [26]. Nitrate nitrogen (NO$_3^-$-N) was determined through Ultraviolet spectrophotometry [27].

### 2.5. Statistical Analysis

The retention formula was used to calculate the removal rate of nitrogen and phosphorus in the GBS as follows:

$$R(\%) = \frac{(C_0 - C)}{C_0} \times 100$$

where $R$ is the removal rate, $C_0$ is the inflow concentration of nitrogen and phosphorus (mg/L), and $C$ is the concentration of nitrogen and phosphorus (mg/L) at each width along the GBS.

Microsoft Excel 2021 (Microsoft Corp. Redmond, DC, USA) and SPSS 19.0 (IBM, Armonk, NY, USA) were used to organize and analyze the experiment data. Graphics were drawn using Origin 2018 (OriginLAb, Northampton, MA, USA).

## 3. Results and Discussions

### 3.1. Characteristic Analysis of the Removal Effect of the GBS on Nitrogen and Phosphorus

#### 3.1.1. TN

TN concentrations for S1, S2, S2 and S4 at different GBS widths ranged from 1.64~4.91 mg/L, 2.10~5.20 mg/L, 1.62~3.68 mg/L, and 3.10~7.48 mg/L (Table 3). Correspondingly, the removal rate was around 29.8~76.5%, 25.7~70.0%, 47.4~76.8%, and 37.6~73.4% (Figure 1). The intercept rate of the contaminant was maximum at a buffer width of 45 m for S2, S3, and S4. However, for S1, the runoff was not collected at a width of 45 m due to the small runoff volume, so its maximum removal rate appeared at a width of 35 m.

**Table 3.** Variation characteristics of contaminant concentration in the runoff water of GBSs.

| Treatment | Width m | TN | $NO_3^--N$ | $NH_4^+-N$ | TP |
| --- | --- | --- | --- | --- | --- |
| | | | mg/L | | |
| S1 | 5 | 4.91 ± 0.88 | 1.92 ± 0.21 | 3.42 ± 0.44 | 0.31 ± 0.06 |
| | 15 | 3.55 ± 0.45 | 1.59 ± 0.34 | 2.00 ± 0.53 | 0.28 ± 0.06 |
| | 25 | 2.66 ± 0.53 | 1.35 ± 0.28 | 1.40 ± 0.38 | 0.29 ± 0.03 |
| | 35 | 1.64 ± 0.56 | 1.16 ± 0.25 | 1.12 ± 0.39 | 0.26 ± 0.06 |
| | 45 | - | - | - | - |
| S2 | 5 | 5.20 ± 0.27 | 1.75 ± 0.13 | 3.70 ± 0.33 | 0.36 ± 0.05 |
| | 15 | 4.25 ± 0.85 | 1.71 ± 0.14 | 2.48 ± 0.39 | 0.32 ± 0.05 |
| | 25 | 2.69 ± 0.70 | 1.67 ± 0.29 | 1.35 ± 0.15 | 0.25 ± 0.09 |
| | 35 | 2.21 ± 0.70 | 1.41 ± 0.26 | 0.80 ± 0.14 | 0.25 ± 0.08 |
| | 45 | 2.10 ± 0.69 | 1.06 ± 0.43 | 0.71 ± 0.28 | 0.22 ± 0.07 |
| S3 | 5 | 3.68 ± 0.74 | 1.49 ± 0.26 | 2.27 ± 0.84 | 0.30 ± 0.05 |
| | 15 | 3.64 ± 0.46 | 1.51 ± 0.24 | 1.89 ± 0.26 | 0.31 ± 0.05 |
| | 25 | 2.93 ± 0.67 | 1.31 ± 0.3 | 1.43 ± 0.42 | 0.26 ± 0.03 |
| | 35 | 2.44 ± 0.56 | 1.24 ± 0.27 | 1.10 ± 0.29 | 0.20 ± 0.05 |
| | 45 | 1.62 ± 0.40 | 1.09 ± 0.25 | 0.49 ± 0.27 | 0.17 ± 0.05 |
| S4 | 5 | 7.48 ± 1.33 | 2.12 ± 0.43 | 5.17 ± 1.08 | 0.63 ± 0.26 |
| | 15 | 5.79 ± 1.43 | 2.23 ± 0.80 | 3.90 ± 0.80 | 0.61 ± 0.21 |
| | 25 | 3.67 ± 0.68 | 1.65 ± 0.43 | 2.22 ± 0.63 | 0.39 ± 0.19 |
| | 35 | 3.35 ± 0.83 | 1.51 ± 0.38 | 1.31 ± 0.25 | 0.37 ± 0.11 |
| | 45 | 3.19 ± 0.30 | 1.47 ± 0.20 | 1.34 ± 0.28 | 0.35 ± 0.07 |

According to S1 and S3, it was found that, under the same inflow velocity and contaminant concentration, the maximum removal rates of runoff TN were similar (76.5% and 76.8%, respectively). For S2 and S3, under the same inflow volume and concentration, the runoff removal rate decreased with the increase in inflow velocity. The maximum interception rate for S2 was only 68.9%. Additionally, with regard to S3 and S4, under the same inflow volume and velocity, the maximum TN removal rate of S4 (68.2%) with higher inflow contaminant concentration was also lower than S3.

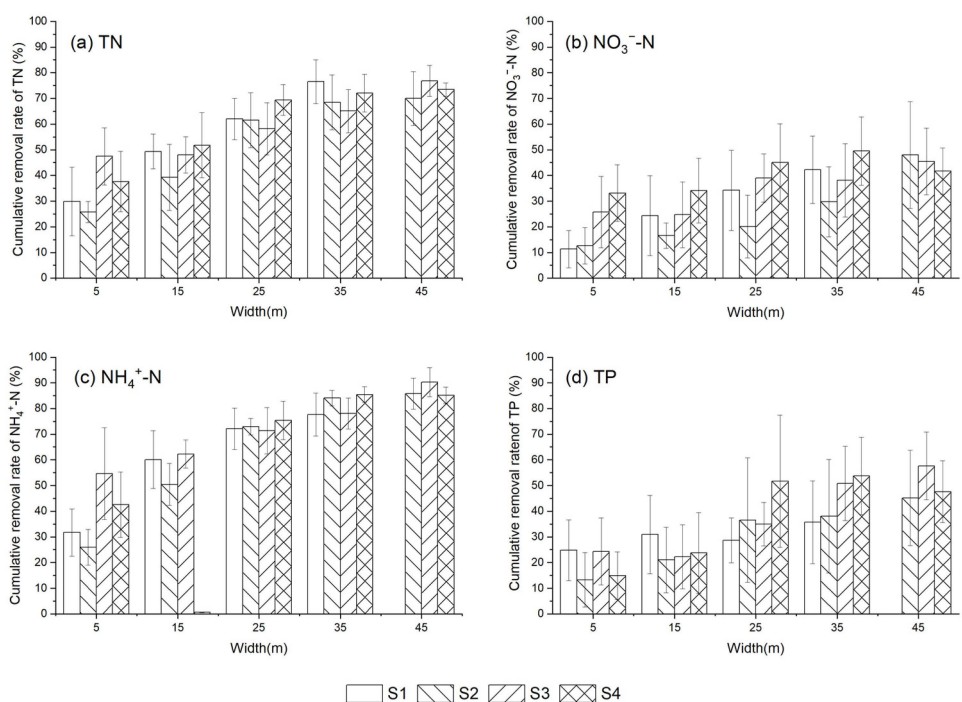

**Figure 1.** Cumulative removal rate of nitrogen and phosphorus from runoff by GBSs.

### 3.1.2. $NO_3^-$-N

$NO_3^-$-N concentrations for S1, S2, S2 and S4 at different GBS widths ranged from 1.16~1.92 mg/L, 1.06~1.75 mg/L, 1.09~1.49 mg/L, and 1.47~2.23 mg/L (Table 3). Correspondingly, the removal rate was around 3.9~42.2%, 12.6~47.2%, 25.7~45.4%, and 29.4~49.5% (Figure 1).

According to S1 and S3, it is found that, under the same inflow velocity and concentration, the maximum removal rate of runoff $NO_3^-$-N were similar (42.2% and 45.4%, respectively). For S2 and S3, under the same inflow volume and concentration, the runoff removal rate slightly improved with the increase in inflow velocity. The maximum removal rate for S2 was 47.2%. Additionally, with regard to S3 and S4, under the same inflow volume and velocity, the maximum $NO_3^-$-N removal rate of S4 (49.5%) with higher inflow contaminant concentration was also higher than S3.

### 3.1.3. $NH_4^+$-N

$NH_4^+$-N concentrations for S1, S2, S2 and S4 at different GBS widths ranged from 1.12~3.42 mg/L, 0.71~3.70 mg/L, 0.49~2.27 mg/L, and 1.34~5.17 mg/L (Table 3). Correspondingly, the removal rate was around 31.7~77.6%, 25.9~85.8%, 54.6~90.3%, and 42.5~85.4% (Figure 1).

According to S1 and S3, it is found that, under the same inflow velocity and concentration, the maximum removal rate of runoff $NH_4^+$-N with higher inflow volume was higher (77.6% vs. 90.3). For S2 and S3, under the same inflow volume and concentration, the runoff removal rate slightly decreased with the increase in inflow velocity. The maximum removal rate for S2 was 85.8%. Additionally, with regard to S3 and S4, under the same inflow volume and velocity, the maximum $NH_4^+$-N removal rate of S4 (85.1%) with higher inflow contaminant concentration was also lower than S3.

### 3.1.4. TP

TP concentrations for S1, S2, S2 and S4 at different GBS widths ranged from 0.26~0.31 mg/L, 0.22~0.36 mg/L, 0.17~0.30 mg/L, and 0.35~0.63 mg/L (Table 3). Correspondingly, the interception rate was around 21.4~35.7%, 10.6~45.1%, 22.2~57.6%, and 21.0~53.7% (Figure 1).

According to S1 and S3, it was found that, under the same inflow velocity and concentration, the maximum removal rate of runoff TP with higher inflow volume was higher (35.7% vs. 57.6%). For S2 and S3, under the same inflow volume and concentration, the runoff removal rate slightly decreased with the increase in inflow velocity. The maximum removal rate for S2 was 45.1%. Additionally, with regard to S3 and S4, under the same inflow volume and velocity, the maximum TP removal rate for S4 (53.7%) with higher inflow contaminant concentration was also lower than for S3.

*3.2. Analysis of the Effect of the GBS Width with Different Inflow Conditions on the Interception of Nitrogen and Phosphorus in Runoff*

### 3.2.1. Inflow Volume

The inflow volume can affect the pollutant removal efficiency of GBSs by affecting the passing time of runoff through the GBS [14,16]. Based on the ANOVA analysis, the inflow volume and the GBS widths were taken as the main influencing factors to analyze their effect on the contamination removal rate. Results showed that the GBS width significantly affects the removal rate of TN, $NO_3^-$-N, $NH_4^+$-N, and TP ($p < 0.01$), and the removal rate increased with the increase of GBS width. Although the interception rate increased with the inflow volume, the inflow volume greatly affected $NH_4^+$-N interception ($p < 0.01$). It was noted that the inflow volume coupled with width significantly affected TN, $NO_3^-$-N, and $NH_4^+$-N ($p < 0.01$). Neither the inflow volume nor inflow volume coupled with GBS widths significantly affected TP interception ($p < 0.05$). Li et al. [28] studied the effect of rainstorm duration on pollutant removal by the buffer zone. It has been found that the removal rate of $NH_4^+$-N under the 2 h rainstorm was better than that of the 0.5 h rainstorm [28]. However, it barely affected the TP interception, similar to the results of this study. In this study, the influence of inflow volume on the removal rate of $NH_4^+$-N was significant ($p < 0.01$), which was related to the gentler slope of the experiment field. When other inflow conditions are determined, the larger the inflow volume, the longer the runoff stays in the GBS, resulting in a higher rate of nitrogen interception.

### 3.2.2. Inflow Velocity

As with the inflow volume, the inflow velocity similarly affected the runoff passing time through the GBS, which then affected the pollutant removal rate. The inflow velocities in this study were 0.74 L/s and 1.48 L/s. For the same runoff volume, the passing time of high-speed runoff over the GBS was reduced by half. Results showed that inflow velocity significantly affected the removal rate of TN, $NO_3^-$-N, and $NH_4^+$-N ($p < 0.05$), indicating that the buffer zone's ability to remove pollutants decreases as the inflow velocity increases. The ability of the buffer zone to remove TP also decreases with the increase in inflow velocity, and the influence was close to the significance level ($p = 0.06$). GBS widths significantly affect the interception of TN, $NO_3^-$-N, and $NH_4^+$-N, and TP ($p < 0.01$). However, the inflow volume coupled with GBS width significantly affected the removal ability of nitrogen ($p < 0.05$), while the ability to intercept TP was not significant ($p > 0.05$). She et al. [20] found that the smaller the inflow velocity, the better the retarding effect of the simulated plant buffer zone, which is similar to the results of this study. In addition, regarding inflow velocity, the removal efficiency of TN, $NH_4^+$-N, and TP by the Bermuda grass buffer zone is also negatively correlated with inflow volume [21]. This indicates that the inflow velocity designed in this study has a more significant impact on the small width of the GBS. With the inflow velocity increase, the buffer zone's ability to intercept nitrogen and phosphorus will decrease, which is consistent with the results of most studies [20,29].

### 3.2.3. Inflow Concentration

While the inflow velocity and inflow volume were similar, the removal rate of contaminants decreased with the increase in inflow concentration (Table 3), and it significantly affected the removal rate of $NH_4^+$-N ($p < 0.05$). The GBS widths significantly affected the removal rate of TN, $NO_3^-$-N, $NH_4^+$-N, and TP ($p < 0.01$). The inflow concentration coupled

with GBS width also significantly affected the interception of TN and $NH_4^+$-N ($p < 0.05$), while it did not reach the significant level for the interception rate of TP ($p < 0.05$). The ability of runoff pollutants to be removed by GBSs is not only related to inflow concentration but also closely related to the form of the pollutant [14]. Nitrogen in runoff simulated in this study exists in the dissolved state. Due to the short runoff erosion times, the ability of herbaceous plants to absorb and transform runoff nitrogen is limited, and the main pathways affecting nitrogen and phosphorus are the adsorption, complexation of soil particles, and runoff infiltration. In addition, the buffer zone is limited in its ability to remove dissolved phosphorus [14,22]. Overall, the buffer zone has a good effect on the removal of $NH_4^+$ mainly because the soil colloid is negatively charged, and the dissolved $NH_4^+$ exists in the state of $NH_4^+$ cation in runoff water, which is easily absorbed by soil colloid and is less easily lost with runoff. Nitrates and phosphates are present in the ionic forms $NO_3^-$ and $PO_4^{3-}$, which are easily dissolved in water and are difficult to absorb by colloids [14,29]. In addition, $NO_3^-$ and $PO_4^{3-}$ are easily absorbed and lost through particulate matter or silt in runoff waters [29,30], but these factors were not considered in this study. Therefore, the $NO_3^-$ and TP removal rates were lower, and the runoff water concentration increased with the width increase in some periods.

### 3.3. Analysis of the Maximum Cumulative Removal Rates for Nitrogen and Phosphorus by GBS in High-Frequency Rainfall Simulations

Under high-frequency rainfall simulation, the maximum cumulative removal rate of TN in runoff ranged from 71.3% to 80.9% (average 75.8%) (Figure 2). The peak of TN removal rate mainly occurred at widths of 35 m and 45 m, with frequencies of 50% and 47%, respectively. The removal rate of $NO_3^-$-N was 34.8%~64.9% (average 50.5%), and the peak of $NO_3^-$-N removal rate similarly appeared at widths of 35 m and 45 m, with the frequency being 44% and 44%, respectively (Figure 2). Additionally, the peak removal rate for $NO_3^-$-N also occurred at GBS widths of 15 m and 25 m, with frequencies of 6% and 12% (Figure 2). For $NH_4^+$-N, the maximum removal rate was 82.3~90.4% (average 85.7%), and the peak of $NH_4^+$-N removal rate mainly occurred at widths of 25 m and 45 m, with frequencies of 44% and 47%, respectively. In addition, the maximum removal rate of TP was 41.8~68.9% (average 51.2%), and the peak frequency of TP removal rate appeared at 25 m, 35 m, and 45 m GBS width, which were 29%, 29%, and 47%, respectively (Figure 2).

The peak removal rate of runoff nitrogen and phosphorus in the GBSs did not decrease with the increase in rainfall frequency. On the contrary, with the increase in rainfall frequency, the peak removal rate of $NH_4^+$-N showed a significant increase trend (y = 0.874x + 80.407, $r^2$ = 0.685) (Figure 2). With the increase in rainfall frequency, the changing trend of cumulative removal rate for TN, $NO_3^-$-N, and TP in GBS was unnoticeable. The peak interception rate of $NO_3^-$-N was slightly low (34.8%) in the 10th rainfall simulation. Besides that, from the sixth rainfall simulation, the peak cumulative removal rates of TN, $NO_3^-$-N, and TP in the GBS were stable at 75%, 55%, and 50%. It was noted that for $NO_3^-$-N, the frequency of peak cumulative removal rate at 15 m and 25 m was 17.6% and 29.4%, respectively. This is probably due to the negative charge of the soil colloids. $NO_3^-$-N and phosphate exist in the ionic form of $NO_3^-$ and $PO_4^{3-}$, which are difficultly adsorbed by colloids [14,22]. In addition, the conversion of $NH_4^+$-N to $NO_3^-$-N via nitrification is also a reason for the decrease in $NO_3^-$-N intercepts as the runoff width increases [14,31,32]. In this study, to avoid the effects of natural rainfall, the experiment was conducted in the fall, when there was little rain. With the increase in simulation times, the temperature in the simulated area decreased. A decrease in temperature may result in a reduction in the amount of $NH_4^+$-N converted through nitrification [13,33], which leads to an increase in $NH_4^+$-N concentration in runoff, and the amount of $NH_4^+$-N removed by the GBS increased. Moreover, the overall cumulative removal rates of $NO_3^-$-N and TP in this experiment were lower, which may be due to the fact that the runoff water configuration in this study does not take into account the presence of large amounts of suspended solids in the water, so the removal effect of nitrogen and phosphorus with high solubility is low [14,34]. Dissolved

nitrogen and phosphorus will adhere to SS, and the removal of nitrogen and phosphorus through removing suspended solids is also a way to utilize the interception ability of the GBS [14,22].

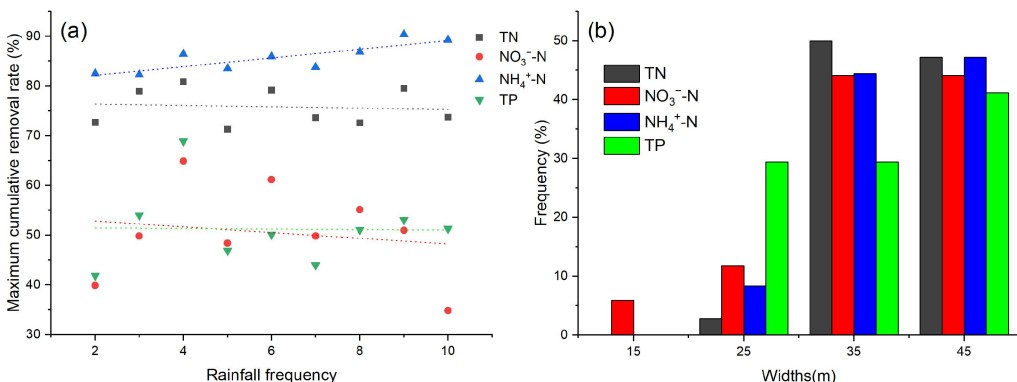

**Figure 2.** Nitrogen and phosphorus cumulative removal rate by GBS in high-frequency rainfall simulation: (**a**) the maximum cumulative removal rate (**b**) width frequency analysis of peak cumulative removal rate.

### 3.4. Analysis of the Optimal Width for Nitrogen and Phosphorus Removal in Paddy Runoff by GBS

The increase in the cumulative removal rate for nitrogen and phosphorus at different GBS widths is shown in Figure 3. The maximum increase rate of cumulative removal rates for TN, $NH_4^+$-N, $NO_3^-$-N, and TP occurred at 0–5 m buffer widths, with values of 35.2%, 18.0%, 38.7%, and 19.4%, respectively. The minimum increase rate of cumulative removal rates occurred at a buffer width of 35–45 m. For TN, the cumulative removal rate increased by more than 10% from 0 to 5 m, 5 to 15 m, and 15 to 25 m widths, but only by 7.8% and 4.9% of growth rate from 25 to 35 m and 35 to 45 m widths. The overall trends for $NH_4^+$-N and $NO_3^-$-N removal were similar to those of TN. For TP, there were two higher increase rates of the cumulative removal rate at the 0–5 m and 15–25 m widths (19.4% and 13.1%, respectively), while the growth rates at the 5–15 m, 15–25 m, and 25–35 m widths were only 2.3% to 6.6%. Overall, the increase in the removal rates for nitrogen and phosphorus from 0 to 25 m width was higher than that from 25 m to 45 m.

The width of the GBS obviously affects its ability to remove runoff pollutants. Shirley and Smith [35] indicated that pollutants in runoff could be effectively removed when the buffer width was between 4.8 m and 49.2 m, while a wider buffer zone only increased the storage space of various nutrients and did not have a significant effect on the absorption and conversion efficiency of nutrients. Gharabaghi et al. [36] found that the reduction efficiency of the buffer zone on suspended matter increased from 50% to 98% when the buffer width increased from 2.5 m to 19.5 m. Additionally, some studies have confirmed that the optimal width is between 10 and 15 m, which could remove 70~90% of the nutrient contaminants [29]. In this study, regardless of inflow conditions, increasing the GBS width significantly improved the removal rate of runoff pollutants ($p < 0.05$). Through fitting the buffer zone width and the average cumulative removal rate, it was found to be consistent with a growing trend as a function of the logarithm, with excellent fits. The variation $r^2$ ranges from 0.54203 to 0.99733.

In addition, the analysis of the fitted equation for the removal rate shows that the peak change point of the removal rate occurred at a width of 15 m for $NO_3^-$-N, TP, and at 25 m width for TN, $NH_4^+$-N (Figure 4). Additionally, the increase rate of the cumulative removal rate increased slowly after the change point (Figures 3 and 4). Within the width of 25 m, the cumulative removal rate of TN, $NO_3^-$-N, $NH_4^+$-N, and TP were 58.2~69.4%, 20.1~45.1%, 71.4~75.3%, and 20.1~45.1%, respectively. Although the peak cumulative removal rate of 70.6~97.2% appeared at the GBS width of 35 m and 45 m, considering the large population density and insufficient land resources in this basin, 25 m was recommended as the optimum width to remove runoff pollutants.

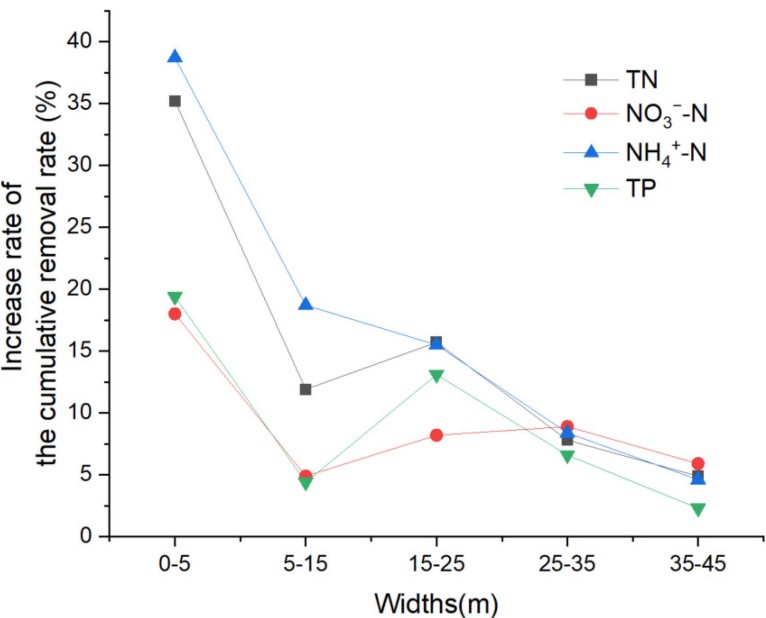

**Figure 3.** Analysis of the increase rate for nitrogen and phosphorus cumulative removal rate at different widths of the GBS.

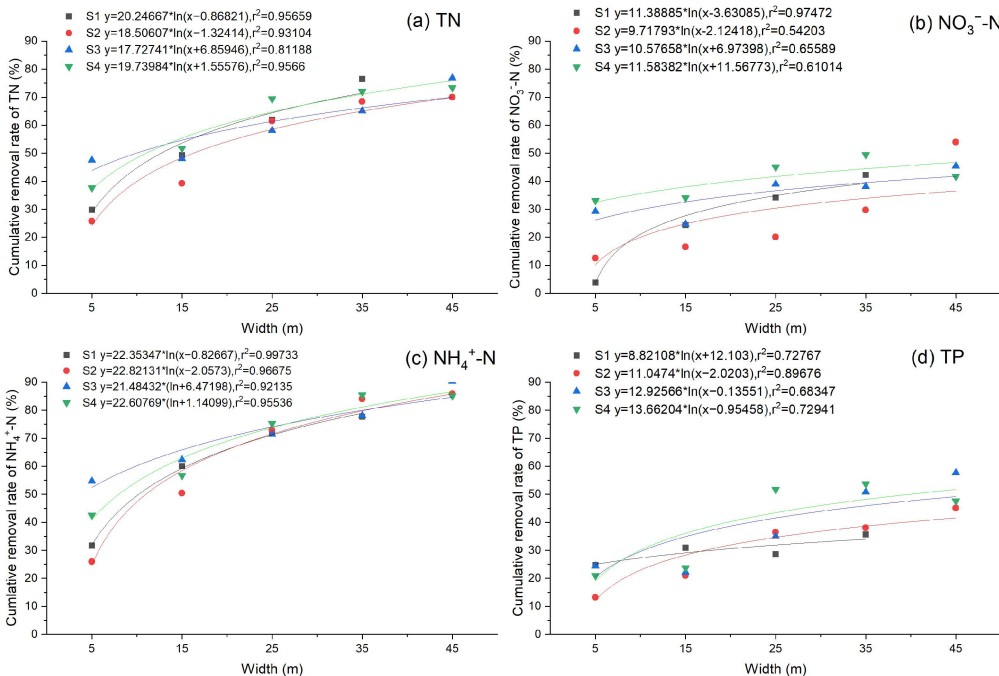

**Figure 4.** Equations fitted to the cumulative removal rates and GBS width.

## 4. Conclusions

This study simulated the effect of GBSs on runoff nitrogen and phosphorus removal efficiency for different inflow volumes, inflow velocities, and inflow concentrations by constructing a field GBS simulation experiment. Results demonstrate that the larger the inflow volume, the faster the inflow velocity, and the lower the inflow concentration, the higher the runoff pollutant removal rate that occurred in the GBSs, and the pollutant interception rate in the buffer zone increased significantly with the increasing width of GBSs.

Moreover, the cumulative removal rate and GBS width were well-fitted through a logarithmic relation, with $r^2$ ranging from 0.54203 to 0.99733. The peak change point of the removal rate occurred at a width of 15 m for $NO_3^--N$, TP, and at a 25 m width for TN,

$NH_4^+$-N. Additionally, the increase rate of the cumulative removal rate increased slowly after the change point. Within the width of 25 m, the removal rate of TN, $NO_3^-$-N, $NH_4^+$-N, and TP were 58.2~69.4%, 20.1~45.1%, 71.4~75.3%, and 20.1~45.1%, respectively. Although the peak cumulative removal rate of 70.6~97.2% appeared at the GBS width of 35 m and 45 m, considering the large population density and insufficient land resources in this basin, 25 m was recommended as the optimum width to remove runoff pollutants.

Thus, this study provides a scientific basis for setting the GBS widths in Taihu Lake Basin. It is worth mentioning that only a single vegetation species was considered, and the silt particles in the runoff were not considered, which provides a direction for further research.

**Author Contributions:** K.M.: Formal analysis, Data Curation, Writing—Original Draft; W.D.: Formal analysis, Data Curation, Investigation, Writing—Original Draft; Z.X.: Methodology, Data Curation, Writing—Review & Editing; C.L.: Validation, Resources, Writing—Review & Editing, Supervision; C.Y.: Methodology, Resources, Writing—Review & Editing, Supervision. All authors have read and agreed to the published version of the manuscript.

**Funding:** This project was supported by the Open Research Fund of the State Environmental Protection Key Laboratory for Lake Pollution Control (2022HPYB-12) and the Fundamental Research Funds for the Central Public-interest Scientific Institution (2022HPYB-12).

**Informed Consent Statement:** Not applicable.

**Data Availability Statement:** The datasets used and/or analyzed during the current study are available from the corresponding author upon reasonable request.

**Conflicts of Interest:** The authors declare no conflict of interest.

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
