# Peer review of "Effect of Grass Buffer Strips on Nitrogen and Phosphorus Removal from Paddy Runoff and Its Optimum Widths"

_agronomy, doi:10.3390/agronomy13122980_

Round 1

Reviewer 1 Report

Comments and Suggestions for Authors

Please refer to the pdf file for comments for improvement

Comments on the Quality of English Language

Minor correction and proofreading is required

Reviewer 2 Report

Comments and Suggestions for Authors

Dear Editor,

Thank you for considering me as a reviewer for the article titled "Research on the Effect of Sward Buffer Zone on Nitrogen and Phosphorus Removal from Paddy Runoff and Its Optimum Widths." My observations are as follows:

·         The title is somewhat long and could be more concise. It's generally advisable to keep titles as brief as possible while still conveying the essence of the research, the term “sward buffer” may be unfamiliar to some readers. Please explain better.

·         Line 22-23 Please rephrase for more clarity.

·         Abstract lacks essential details such as the location of the study or the specific pollutants examined. Including these details in the abstract would provide context and make the research more accessible to a wider audience.

·         L31-33. Please rephrase for clarity, such as starting with a general discussion of non-point source pollution and then gradually narrowing down to the specific issue of AGNPS in the Taihu Lake basin. This will help readers better understand the context and importance of your research.

·         You use "AGNPS" initially, but later refer to it as "AGNPS pollution." Choose one consistent way to refer to it to avoid confusion.

·         L34-37 When discussing the contribution of AGNPS pollution to the nitrogen and phosphorus input in the Yangtze River, you could compare it to other sources of pollution or historic data.

·         L49-50. Could you please explain the mechanism for this?

·         The inconsistent use of terminology can be confusing. For example, AGNPS is referred to as both "AGNPS" and "AGNPS pollution." Maintain consistent terminology throughout the introduction.

·         While you cite various studies and data sources, it's necessary to briefly discuss the reliability and relevance of these sources. Readers should know the quality and context of the data you're relying on.

·         Some sentences are overly complex, making the text less reader-friendly. Simplify sentence structure to ensure clarity and comprehension. For example, L47-55, L60-74.

·         When discussing high-frequency rainfall, provide a concise explanation of why it's relevant in the context of AGNPS pollution in the Taihu Basin. What distinguishes high-frequency rainfall and why is it important to your study?

·         The introduction doesn't clearly state the research objectives or questions that the study intends to address. It is essential to outline the specific goals to provide a sense of direction for the readers.

·         It is better to separate the results and the discussion to discuss holistically the results rather than results-discussion together. I believe the discussion part need more depth to explore and relate this article data findings. This is not a review article.....please discuss your results deeply. I would suggest at First author should enlist most eminent findings of his results and then discuss one by one. The presented discussion part is very weak.

·         In conclusion, you recommend a buffer width of 15m-25m, the basis for this recommendation is not clearly explained. What factors led to this specific recommendation, and how does it relate to the study's objectives and broader? A clear connection between what was initially sought and what was actually achieved should be established.

·         It's important to acknowledge the limitations of your study and suggest directions for future research. Mentioning what this study did not cover or where further investigation is needed can strengthen the conclusion.

Comments on the Quality of English Language

Minor editing of English language required
